# Mechanism and Applications of Vagus Nerve Stimulation

**DOI:** 10.3390/cimb47020122

**Published:** 2025-02-14

**Authors:** Zhen Chen, Kezhou Liu

**Affiliations:** Department of Biomedical Engineering, School of Automation (Artificial Intelligence), Hangzhou Dianzi University, Hangzhou 310018, China; chenzhendj@icloud.com

**Keywords:** vagus nerve stimulation, neuromodulation, mechanism, clinical applications

## Abstract

Over the past three decades, vagus nerve stimulation (VNS) has emerged as a promising rehabilitation therapy for a diverse range of conditions, demonstrating substantial clinical potential. This review summarizes the in vivo biological mechanisms activated by VNS and their corresponding clinical applications. Furthermore, it outlines the selection of parameters and equipment for VNS implementation. VNS exhibits anti-inflammatory effects, modulates neurotransmitter release, enhances neural plasticity, inhibits apoptosis and autophagy, maintains blood–brain barrier integrity, and promotes angiogenesis. Clinically, VNS has been utilized in the treatment of epilepsy, depression, headache, stroke, and obesity. Its potential applications extend to anti-inflammatory treatment and the management of cardiovascular and cerebrovascular diseases and various brain disorders. However, further experiments are required to definitively establish the efficacy of VNS’s various mechanisms. Additionally, there is a need to explore and identify optimal rehabilitation treatment parameters for different diseases.

## 1. Introduction

The vagus nerve (VN) is a major component of the parasympathetic nervous system, which can transmit various stimuli related to bodily sensations, including pressure, pain, stretching, temperature, chemicals, osmotic pressure, and inflammation [1]. The VN emerges from the medulla oblongata, traverses the jugular foramen to exit the cranial cavity, and descends within the neurovascular bundle between the internal jugular vein and the common carotid artery. It subsequently extends into the thoracic and abdominal cavities, where it extensively innervates multiple visceral organs, including the heart, lungs, and gastrointestinal tract [2]. Notably, the auricular branch of the vagus nerve (ABVN) represents the sole superficial afferent component of the VN system, providing sensory innervation to the external acoustic meatus, inner tragus, and the periauricular skin surrounding the cymba conchae [3]. Importantly, the cymba conchae region receives exclusive neural supply from the ABVN [4]. The VN plays a critical role in the modulation of cardiac function, gastrointestinal motility, respiratory regulation, and immune system homeostasis [5].

In the late 19th century, inspired by electrotherapy, James Corning proposed the concept of vagus nerve stimulation (VNS) and attempted to treat epilepsy by modulating neuronal excitability through electrical stimulation of the VN. He hypothesized that such stimulation might suppress abnormal brain activity, reducing seizure frequency and severity [6]. Although his efforts lacked clinical success, subsequent animal studies, including Zabara et al.’s work [7], demonstrated VNS’s potential by interrupting seizures and prolonging suppression in canine epilepsy models. In 1997, the Food and Drug Administration (FDA) approved the first implantable VNS device for the treatment of epilepsy. In 1988, the world’s first VNS surgery was performed for the intervention of epilepsy [8]. Since then, VNS has been approved for the treatment of various diseases.

In this review, we investigated the potential of VNS and the possible connections between its clinical applications. Additionally, we summarized the issues related to parameter selection when using VNS.

## 2. The Potential Mechanism of VNS

VNS plays its role by stimulating the afferent and efferent fibers of the VN. These fibers project upward to the brainstem nucleus and relay circuit, influencing the nucleus tractus solitarius (NTS) and locus coeruleus (LC), and downward to the internal organs, affecting the autonomic, neuroendocrine, and neuroimmune systems [9]. The following are the related molecular mechanisms triggered by VNS.

### 2.1. Anti-Inflammation

The cholinergic activity of the efferent branch of the vagus nerve has an immune–inflammatory regulatory effect, which is referred to as the cholinergic anti-inflammatory pathway (CAP). VNS exerts its anti-inflammatory effects through CAP. Specifically, the efferent branch of the vagus nerve releases acetylcholine (ACh), which activates α7nAChR receptors on immune cells, initiating the Jak2-STAT3 pathway [10]. This leads to changes in the expression levels of cytokines (e.g., TNF-α, IL-6, and IL-1β) associated with inflammation.

The activation of α7nAChR is a necessary condition for exerting the anti-inflammatory effects in this process [11]. VNS is believed to activate α7nAChR through multiple mechanisms (Figure 1a). Firstly, VNS can directly activate α7nAChR in resident macrophages in the small intestine through the intestinal VN neurons [12]. Secondly, action potentials originating from the VN can regulate the release of norepinephrine (NE), and then, NE binds to memory T cells that can produce ACh, thereby allowing T cells to produce the neurotransmitter ACh required to control innate immune responses [13]. ACh produced by these T cells ultimately activates α7nAChR on splenic macrophages [14]. Finally, VNS can activate α7nAChR on microglial cells, thereby alleviating neuroinflammation in the brain [15].

The exact anti-inflammatory mechanisms of VNS are not yet fully understood. Apart from CAP, VNS may also regulate inflammation through other means. One study found that VNS enhances the expression of peroxisome proliferator-activated receptor gamma (PPAR-γ) and inhibits the expression of pro-inflammatory cytokines and immune cell activation in ischemic penumbra, suggesting that PPAR-γ may be involved in the VNS-induced anti-inflammatory process [16]. VNS may also influence the hypothalamic–pituitary–adrenal axis to regulate inflammation [17], though more research is needed to provide evidence for these viewpoints.

### 2.2. Regulation of the Release of Neurotransmitters

VNS can regulate the release of various neurotransmitters to achieve related physiological effects (Figure 1f). In addition to stimulating the release of Ach and initiating the CAP, VNS can also regulate the release of monoaminergic neurotransmitters and gamma-aminobutyric acid (GABA) energic neurotransmitters to exert neuroprotective effects.

Monoaminergic neurotransmitters, including norepinephrine (NE), serotonin (5-HT), and dopamine (DA), play a role in the treatment of brain disorders such as epilepsy and depression. VNS can increase the release of NE in the locus coeruleus [18] and basolateral amygdala [19], leading to increased extracellular NE levels in the prefrontal cortex and hippocampus. Chronic VNS significantly increases extracellular 5-HT levels in the dorsal raphe nucleus, but not in the hippocampus or prefrontal cortex. Electrophysiological experiments show that long-term VNS reduces the firing rate of ventral tegmental area DA neurons, while increasing extracellular DA levels in the prefrontal cortex and septum [20].

GABA is an inhibitory neurotransmitter associated with various brain disorders [21]. One study showed that VNS treatment in patients with epilepsy significantly increases the levels of total free GABA in the body [22]. VNS can regulate cortical excitability in brain regions associated with epilepsy, and GABA(A) receptor plasticity contributes to this effect [23]. GABA(A) and GABA(B) receptors are involved in the central mechanisms regulating VNS-induced theta oscillations in the hippocampus [24]. These results suggest that VNS may act through the GABAergic system to exert its effects in brain disorders.

### 2.3. Neuroplasticity

Plasticity is crucial for functional recovery when brain tissue undergoes trauma or damage caused by ischemia [25]. Brain-derived neurotrophic factor (BDNF) is an important regulator of hippocampal plasticity and neurogenesis that may influence the formation of learning and memory [26]. One month of VNS treatment increased the expression of BDNF and dendritic complexity in the rat hippocampus [27], and enhanced neuronal excitability in the hippocampal CA1 region was observed two weeks after VNS treatment [28]. VNS can rapidly activate phosphorylation of the BDNF receptor TrkB, and this effect persists over time [29]. Similarly, a recent study reported that taVNS enhances axonal plasticity and improves long-term neurological recovery by activating the BDNF signaling pathway via α7nAChR [30]. These studies suggest that VNS-induced neuroplasticity is likely associated with the activation of the BDNF-TrkB pathway (Figure 1d).

Researchers have found that the cholinergic nucleus basalis is essential for VNS-dependent enhancement of plasticity in the motor cortex [31]. It has also been found that the VNS effect disappears when pharmacological depletion of noradrenergic, serotonergic, or cholinergic neurotransmission occurs in rats [32]. These findings indicate that VNS can promote the formation of a neurochemical environment conducive to synaptic plasticity in the lesion area.

Neuroplasticity induced by VNS is not a spontaneous process but is highly dependent on the pairing of VNS with specific rehabilitative training. The concurrent application of VNS during rehabilitation enhances synaptic plasticity, strengthens neural circuits, and promotes functional recovery by leveraging the timing-dependent effects of VNS on neuromodulatory systems. This pairing is essential for optimizing the therapeutic outcomes of VNS, as it facilitates the reorganization of neural networks in response to task-specific training. For example, repeatedly pairing tones with brief pulses of VNS completely eliminated the physiological and behavioral correlates of tinnitus in noise-exposed rats [33].

### 2.4. Inhabition of Cell Apoptosis and Autophagy

VNS significantly reduced the levels of Caspase-3 protein in the ischemic penumbra, suggesting that VNS may exert neuroprotective effects by inhibiting cell apoptosis and autophagy in the lesion area [9]. miR-210, an important microRNA regulated by hypoxia-inducible factor and the Akt-dependent pathway, may be associated with VNS’s anti-apoptotic effects on ischemia/reperfusion (I/R) injury [34]. VNS enhanced the expression of miR-210 in ischemic stroke and reduced the expression levels of Caspase-3 protein associated with cell apoptosis. This effect was significantly weakened after silencing miR-210, and the therapeutic effect of VNS disappeared. Lipocalin prostaglandin D2 synthase (L-PGDS) may be involved in inhibiting VNS-induced apoptosis in I/R injury. Inhibition of L-PGDS expression weakened the anti-apoptotic effect induced by VNS [35].

A recent study reported that compared to the I/R group, VNS significantly downregulated the expression of autophagy-related proteins Beclin-1 and LC3-II, downregulated the expression of pro-apoptotic protein Bax, and upregulated the expression of anti-apoptotic protein Bcl-2, suggesting that VNS may exert neuroprotective effects by inhibiting the cell autophagy pathway [36]. The specific molecular mechanism by which VNS inhibits cell apoptosis and autophagy remains unknown and requires further research.

### 2.5. Regulation of BBB Permeability

The blood–brain barrier (BBB) plays an important role in maintaining the homeostasis of the central nervous system, and VNS may exert neuroprotective effects by reducing BBB permeability. Giving invasive vagus nerve stimulation (iVNS) to mice after traumatic brain injury reduced the permeability of brain’s blood vessels [37]. VNS exerted neuroprotective effects by improving the integrity of the BBB and reducing the activation of microglial cells and astrocytes in mice with small brain infarcts [38]. VNS reduced BBB disruption in a rat model of ischemic stroke, protected tight junction proteins from microvascular damage, and reduced the expression of matrix metalloproteinase-2/9 (MMP-2/9) in activated perivascular astrocytes surrounding the damaged vessels [39]. However, the specific mechanism by which VNS affects the BBB is still unclear. Some speculate that VNS may affect BBB permeability by influencing Ach and NE around the BBB to alleviate neuroinflammation [40].

### 2.6. Promotion of Angiogenesis

Cerebrovascular remodeling and regeneration play crucial roles in the recovery stage of ischemic brain injury. Early on, it was found that VNS reduced the infarct volume of focal brain ischemia in rats [41], and it was speculated that this was a result of VNS affecting cerebral blood flow. A subsequent study negated this claim [42]. In 2016, a study reported that transcutaneous auricular vagus nerve stimulation (taVNS) significantly increased the microvessel density and endothelial cell proliferation around the ischemic infarct area in rats, indicating that taVNS promoted vascular regeneration after ischemia [43]. VNS-mediated angiogenesis after ischemic stroke is related to the expression of angiogenic factors such as vascular endothelial growth factor (VEGF), endothelial nitric oxide synthase (eNOS), BDNF, and growth differentiation factor 11 (GDF-11). VNS increased the expression of VEGF, BDNF, and eNOS in the ischemic penumbra of MCAO rats. Subsequently, a study found that the mechanism of VNS-mediated vascular regeneration may be related to GDF11/ALK5. This study found that taVNS promoted the proliferation of brain capillary endothelial cells and the expression of activin-like kinase 5, upregulated brain GDF11, and downregulated splenic GDF11 [44]. There is limited research on VNS promoting angiogenesis, and further studies are needed to confirm its effects and specific mechanisms.

### 2.7. Others

Although scientists have conducted many studies on the mechanisms of vagus nerve stimulation, its full picture has not been revealed. The mainstream view is that VNS promotes the release of neurotransmitters at the vagus nerve endings, activating anti-inflammatory signaling pathways and resulting in a series of biological outcomes. In addition to the mechanisms summarized above, researchers have also found that VNS can affect cerebral blood flow (CBF), glutamate excitotoxicity, ghrelin, and other factors. In a rat model of ischemic stroke, VNS has been shown to lower CBF during a 30 s stimulation period and reduce brain edema after brain injury [45]. VNS significantly reduced ischemia-induced glutamate release and transiently increased hippocampal blood flow in a gerbil model of ischemia–reperfusion [46]. A preliminary study demonstrated that VNS can significantly increase ghrelin levels after traumatic brain injury [47]. The research on the mechanism of VNS is currently only scratching the surface, and more studies are needed to reveal the truth.

## 3. Clinical and Potential Applications

VNS has garnered widespread attention in both laboratory and clinical settings. Most research has focused on VNS therapy for epilepsy, depression, and stroke, but other therapeutic effects of VNS have also been discovered, such as aiding in weight loss, improving symptoms of cardiovascular disease, and alleviating mood disorders. The following will provide an overview of the approved clinical uses of VNS and potential disease therapies.

### 3.1. Current Clinical Uses of VNS

#### 3.1.1. Epilepsy

The mechanisms underlying VNS for the treatment of epilepsy have not been fully elucidated but may be associated with the following factors: (1) desynchronization of electroencephalogram (EEG) activity through the nucleus tractus solitarius and the medullary reticular formation pathway and (2) reduction in excitatory neurotransmission and enhancement of inhibitory neurotransmission [48].

The initial concept of VNS was proposed for the treatment of epilepsy, but due to technical reasons, it was not successful at the time. In 1997, the first VNS device was approved by the FDA for the treatment of medically refractory partial-onset seizures in patients aged 12 and older. In a randomized double-blind trial, patients receiving high stimulation (94 patients, aged 13 to 54) showed an average reduction of 28% in total seizure frequency, while the low stimulation group (102 patients, aged 15 to 60; *p* = 0.04) showed a 15% reduction [49]. With the development of non-invasive vagus nerve stimulation (nVNS), several treatment studies for refractory epilepsy patients have shown that taVNS can significantly reduce seizure frequency [50]. VNS has now become an effective and safe adjunctive therapy for the clinical treatment of epilepsy, although the exact mechanism by which VNS controls epilepsy remains to be elucidated.

#### 3.1.2. Depression

During studies using VNS to treat epilepsy, it was observed that the mood of the treated patients improved [51], suggesting the potential of VNS for treating depression. The potential mechanisms of VNS in the treatment of depression may include the following: (1) VNS modulates the release of monoamine neurotransmitters, whose dysregulation is closely associated with the pathophysiology of depression; (2) VNS can promote neural plasticity, including increasing the expression of BDNF, thereby improving the function of the hippocampus and prefrontal cortex, with these brain regions playing important roles in emotion regulation and cognitive function [52]; (3) VNS activates the CAP, thereby attenuating neuroinflammation, which is recognized as a critical pathological mechanism underlying depression; and (4) VNS regulates functional connectivity within the default mode network (DMN) and limbic system, enhancing emotional processing and cognitive function, which contributes to the reduction in negative emotional responses [53].

Rush et al. conducted the first study in 2000 investigating the effects of VNS on depression in non-epileptic patients, and the results indicated that VNS had a good antidepressant effect on treatment-resistant depression patients [54]. In 2005, the FDA approved the VNS device manufactured by Cyberonics for the treatment of treatment-resistant depression based on several clinical studies [55,56,57].

#### 3.1.3. Obesity

Inspired by the weight loss in epilepsy patients treated with VNS, metabolic scientists conducted a series of studies on the effects of VNS on weight, food intake, and obesity treatment. The mechanism of VNS treatment for obesity involves multiple physiological processes. (1) Appetite regulation: VNS affects the feeding center of the hypothalamus, thereby regulating the secretion of appetite-related hormones (such as leptin and ghrelin), which helps reduce hunger and increase satiety. (2) Metabolic regulation: VNS can regulate the function of the liver and pancreas, affecting glucose metabolism and insulin sensitivity, which helps improve energy metabolism and reduce fat storage. (3) Intestinal brain axis regulation: VNS regulates gut microbiota and hormone secretion through bidirectional communication between the VN and the gut, thereby affecting appetite and metabolic function [58]. (4) Immune regulation: VNS can affect the release of Ach from the VN endings to activate CAP, and cholinergic mechanisms are associated with reducing obesity-related inflammation and metabolic complications [59]. In 2015, based on the results of the ReCharge trial, the FDA approved the use of the Maestro rechargeable device for the treatment of obesity [60].

#### 3.1.4. Headache

Some case reports have suggested that migraines in treatment-resistant epilepsy patients treated with VNS improved, indicating the potential benefit of VNS for the preventive treatment of migraines [61]. Subsequently, a pilot study demonstrated potential tolerability and effectiveness of nVNS devices in treating migraines [62]. In 2018, nVNS was approved by the FDA for the treatment of migraines, followed by approval for cluster headaches in 2019 [63]. Clinically, taVNS has become an effective non-pharmacological method for treating headaches. The latest clinical studies provide evidence for the safety and efficacy of taVNS in treating cluster headache [64].

The potential mechanisms of VNS in the treatment of headaches, particularly mi-graine and cluster headaches, may include the following. (1) Neuromodulation effects: VNS influences the NTS and LC in the brainstem, thereby modulating the transmission of pain signals. Additionally, VNS suppresses the activity of the trigeminal nucleus caudalis, reducing neuroinflammation and pain perception associated with headaches [65]. (2) Anti-inflammatory effects: VNS activates the CAP, inhibiting the release of pro-inflammatory cytokines, thereby alleviating neuroinflammation related to headaches. (3) Modulation of brain network function: VNS regulates functional connectivity within the DMN and pain-processing brain regions (e.g., anterior cingulate cortex and insula), improving pain perception and emotional regulation in headache patients. (4) Neurotransmitter modulation: VNS modulates the release of monoamine neurotransmitters, influencing pain thresholds and emotional states, which contributes to the relief of headache symptoms.

#### 3.1.5. Stroke

In 2009, team Ay found that VNS reduced the infarct volume in rats with cerebral ischemic stroke, suggesting that VNS may have a positive impact on stroke recovery [66]. In 2013, team Hays found that providing VNS during ischemic stroke recovery training can increase upper limb strength in rats, and they subsequently proposed the paired VNS (or VNS-REHAB) therapy [67]. Paired VNS therapy involves combining VNS with rehab exercises, meaning that VNS is administered while the patient performs rehabilitation exercises. In 2021, the FDA approved paired VNS therapy for clinical trials, and the results indicated that paired VNS therapy is a novel treatment option for patients with impaired upper limbs after ischemic stroke [68].

VNS has emerged as a promising therapy for stroke rehabilitation, with several potential mechanisms underpinning its therapeutic effects. These mechanisms include (1) anti-neuroinflammation, as VNS activates the CAP, reducing the release of pro-inflammatory cytokines such as TNF-α and IL-1β, which are known to exacerbate neuronal damage post-stroke; (2) promoting neural plasticity, as VNS enhances the release of neuromodulators like ACh, NE, and BDNF, which facilitate synaptic reorganization and functional recovery in stroke-affected brain regions; (3) inhibiting cell apoptosis and autophagy, as VNS has been shown to reduce oxidative stress and mitochondrial dysfunction, thereby protecting neurons from programmed cell death; (4) regulation of BBB permeability, as VNS modulates endothelial tight junction proteins, reducing BBB leakage and preventing secondary brain injury; and (5) promoting angiogenesis, as VNS stimulates the release of VEGF, enhancing blood vessel formation and improving cerebral blood flow in ischemic regions. Despite these promising mechanisms, the evidence supporting VNS in stroke rehabilitation remains limited, with only a few clinical and animal studies providing preliminary data. For instance, a randomized controlled trial demonstrated that VNS paired with motor training improved upper limb function in stroke patients [69], while animal studies have shown reduced infarct volume and improved functional outcomes following VNS [67]. However, more rigorous clinical trials and mechanistic studies are needed to validate these findings and optimize VNS protocols for stroke rehabilitation. Future research should focus on elucidating the dose–response relationship, long-term efficacy, and potential side effects of VNS in stroke patients, as well as exploring its synergistic effects with other rehabilitation therapies.

### 3.2. Potential Uses of VNS

#### 3.2.1. Anti-Inflammatory Treatment

A large body of research suggests that VNS can exert anti-inflammatory effects by influencing the expression of inflammatory factors through the cholinergic pathway. Therefore, VNS can be used as a non-pharmacological anti-inflammatory treatment for a variety of diseases such as enteritis, renal inflammation, rheumatoid arthritis, and others.

Inflammatory bowel disease (IBD) is a group of inflammatory diseases involving the colon and small intestine, typically divided into ulcerative colitis and Crohn’s disease. Currently, there is no cure for IBD, and the most common treatment is using anti-TNF drugs, which come with many issues and side effects [70]. Research has shown that VNS improved colitis in rats, demonstrating its anti-inflammatory effects during gastrointestinal inflammation [71,72]. A pilot study on Crohn’s disease showed that VNS restored vagal nerve tone and reduced inflammation in patients [73].

VNS variably affected gene expression in donor organs in kidney transplants, improving renal function in recipients [74,75]. This suggests the potential of VNS in treating kidney inflammation-related diseases. A recent pilot clinical trial involving VNS therapy for hemodialysis patients showed no significant changes in cytokine levels [76], warranting further research to evaluate the efficacy of VNS in hemodialysis patients and other renal pathologies.

Rheumatoid arthritis (RA) is a chronic autoimmune disease, and inhibiting the expression and release of pro-inflammatory cytokines has been a focus of treatment. Research has shown that VNS inhibits the production of TNF-α and reduces the severity of RA [77]. A recent pilot study provided taVNS to 36 high- or low-disease-activity RA patients, suggesting that short-term taVNS may reduce disease activity and pro-inflammatory cytokines, providing preliminary support for the anti-inflammatory effects of VNS in RA patients [78].

Abdominal surgery can induce inflammation of the muscularis externa and bowel paralysis, commonly referred to as postoperative ileus (POI). Additionally, major surgical trauma may trigger systemic inflammation, ranging in severity up to its most severe manifestation, systemic inflammatory response syndrome (SIRS) [79]. The latest human studies support the safety, treatment compliance, and usability of nVNS in patients undergoing major colorectal surgery [80]. A double-blind clinical pilot experiment showed that taVNS was well tolerated in the treatment of SIRS, with little to no side effects [81]. In the future, taVNS will become one of the effective means to treat POI and SIRS, but more ingenious clinical trial models and more convincing data are still needed.

#### 3.2.2. Cardiovascular Disorders

VNS shows a variety of potential mechanisms and application values in the treatment of cardiovascular diseases. Its main mechanisms can be summarized as follows: (1) in terms of autonomic nerve balance regulation, VNS can effectively inhibit the over-excitation of the sympathetic nerve by activating parasympathetic nerve fibers of VN, so as to reconstruct the dynamic balance in the autonomic nervous system; (2) VNS has significant anti-inflammatory effect, where activating the CAP can effectively inhibit the release of pro-inflammatory cytokines, thereby reducing the inflammatory response in cardiovascular disease; (3) VNS can significantly improve cardiac electrophysiological characteristics; (4) VNS showed a clear myocardial protective effect, significantly enhancing the viability of cardiomyocytes by reducing oxidative stress and inhibiting apoptosis, with experimental studies having confirmed that VNS can improve myocardial atrophy in rats [82]; (5) in terms of hemodynamic regulation, VNS can significantly improve the blood supply of the heart by regulating vascular tension and optimizing blood flow distribution; and (6) VNS can effectively reduce the release of stress hormones (such as cortisol) by regulating the hypothalamus pituitary adrenal axis, thereby reducing the stress load of the heart.

Preclinical studies evaluating the use of VNS to promote resuscitation in cardiac arrest rats showed a 90.91% success rate in achieving return of spontaneous circulation when VNS was combined with cardiopulmonary resuscitation (CPR), compared to 83.33% with CPR alone [83]. Additionally, VNS reduced the number of shocks and CPR time required. VNS also improved various post-resuscitation outcomes, including better recovery of cardiac and brain function and longer survival in VNS-treated animals.

Heart failure refers to the inability of the myocardium to maintain sufficient blood flow to meet the body’s needs. Positive effects of VNS on heart failure have been observed in some animal model experiments, making VNS a potential new therapy for treating heart failure. In 2004, a study found that VNS significantly improved the long-term survival rates in rats with chronic heart failure, reducing the relative risk of death by 73% [84]. Another study found that long-term open-loop low-level VNS treatment in dogs with heart failure improved left ventricular contractile function, prevented progressive left ventricular enlargement, and improved heart failure biomarkers [85]. However, in a recent multinational clinical trial, VNS was found not to reduce the mortality or heart failure events in patients with chronic heart failure [86]. These results contradict laboratory data, and more research is needed to confirm the relationship between VNS and heart failure.

Myocardial infarction, commonly known as a heart attack, was found to be effectively improved by taVNS [87], and low-level VNS was found to improve cardiac performance, accompanied by a reversal of dysfunctional calcium handling characteristics [88], potentially indicating a molecular mechanism for VNS in treating heart failure. Immediate application of VNS during myocardial ischemia can prevent I/R injury, and even if started after ischemia, VNS can provide significant cardiac protection, although it is ineffective after reperfusion [89]. These findings highlight the importance of the timing of VNS initiation and provide evidence of VNS’s role in protecting the myocardium from I/R injury risk.

Atrial fibrillation is a relatively common arrhythmia characterized by rapid and irregular beating of the atria and is believed to be influenced by cardiac factors related to sympathetic and parasympathetic nervous system tone. Therefore, research is ongoing on using VNS for atrial fibrillation. Studies have reported the induction of atrial fibrillation in canine models using rapid atrial pacing and its treatment with VNS, which reversed atrial remodeling and reduced the inducibility of atrial fibrillation [90]. Furthermore, VNS was found to suppress atrial fibrillation by shortening its duration and prolonging the atrial fibrillation cycle length [91].

Numerous research results indicate that VNS is beneficial in treating cardiovascular diseases, but it may also lead to severe cardiovascular adverse reactions, such as bradycardia or even cardiac arrest [92]. Therefore, caution must be exercised in human trials. It is hoped that in the future, more researchers will focus on the application of VNS in cardiovascular diseases.

#### 3.2.3. Brain Diseases

VN plays a crucial role in memory, emotion, and pain. The ways in which VNS is used to treat brain diseases may include (1) activating CAP to reduce neuroinflammation; (2) using VNS combined with specific rehabilitation exercise training to promote neural plasticity; (3) regulating the release of related neurotransmitters; and (4) directly controlling autonomic nerve function through the parasympathetic nerve and affecting the excitation state of relevant brain regions. The application of VNS has significantly improved the clinical symptoms of a large number of patients with various brain diseases. Therefore, VNS has been successively approved by the FDA as an alternative treatment for refractory epilepsy, refractory depression, cluster headaches, and migraines. In addition, the use of VNS has been extended to research and treat more brain diseases, such as Parkinson’s disease (PD), autistic spectrum disorder (ASD), Alzheimer’s disease (AD), traumatic brain injury (TBI), and anxiety.

PD is a progressive neurodegenerative disease. Preclinical studies have established a connection between VN and the development of PD [93]. In 2019, scientists applied tcVNS to PD patients for the first time and found that it improved gait and reduced gait freezing in patients with PD [94]. In addition, higher-frequency VNS contributes to greater behavioral improvement and attenuation of pathological markers in PD models [95]. Therefore, VNS may be a potential treatment for PD.

ASD is a lifelong condition characterized by impairments in social communication. Current research has found that taVNS can enhance emotion recognition, which plays a key role in social interaction, thereby reducing autism symptoms [96]. In addition, preliminary clinical studies have found that children with epilepsy and autism experience positive behavioral changes after using VNS [97].

When studying VNS for the treatment of epilepsy, it was found that iVNS can help alleviate anxiety [98]. In addition, taVNS has been shown to accelerate the extinction of conditioned fear, but it is not well retained after 24 h [99]. These data come from a small sample size, so whether taVNS treats anxiety and prevents its recurrence is controversial, and more research is needed.

AD is a neurodegenerative disease which is characterized by neuronal deposition of amyloid-β plaques and neurofibrillary tau tangles, and inflammatory activation of glia [100]. At present, there is no effective treatment, but VNS emerges as a promising intervention for AD, targeting cognitive impairments through the modulation of tau proteins and addressing inflammatory and stress-related pathways through cortisol and telomere regulation. VNS as an intervention in both early and late AD should be investigated more in depth [101].

TBI refers to direct or indirect brain damage caused by external violence. In animal studies, paired VNS therapy has been shown to enhance functional motor recovery after TBI [102]. Clinical studies have found that iVNS increases cerebral blood flow and metabolic activity in the forebrain, thalamus, and reticular formation of patients with moderate to severe TBI, thereby improving physical movement and consciousness in the long term and ultimately improving brain damage [103]. A study reported the first case of a patient with disorders of consciousness after TBI who opened their eyes and cleared their sleep–wake cycle after using taVNS [104]. Recently, animal studies by Collins et al. have also found that VNS can induce widespread cortical activation and increased arousal [105]. These studies suggest that VNS is a promising treatment option for patients with TBI.

#### 3.2.4. Pain, Tinnitus, and Insomnia

Fibromyalgia (FM) syndrome is a common condition characterized by chronic widespread pain, sleep problems, fatigue, and cognitive difficulties. An open-label phase I/II trial suggests that VNS may be a useful adjunctive treatment for FM patients who are resistant to conventional management [106]. Another randomized clinical trial found that taVNS provided pain relief and improved the quality of life for FM patients [107]. These studies lack a sufficient sample size to provide statistical evidence, and future research should increase the sample size and conduct more relevant studies.

Tinnitus is the perception of a phantom auditory sensation in the absence of an external sound source. Animal research has shown that VNS paired with sound stimuli can induce neural plasticity in the auditory cortex in a controlled manner [108]. A few human studies that followed suggested that VNS paired with tones stripped of the tinnitus frequency might improve tinnitus-related symptoms [109,110,111]. There is no reliable evidence to date showing that taVNS alone without paired sound stimuli is effective for tinnitus treatment. Higher-quality research is needed for both paired and unpaired taVNS to compensate for the flaws of existing studies and address the gaps in the current knowledge on the subject, such as proper stimulation parameters, longer follow-up periods, and the most responsive tinnitus populations [112].

TaVNS has been reported to be effective in the treatment of primary insomnia. Primary insomnia patients often show autonomic nervous system imbalance, elevated levels of neurogenic inflammatory factors, and hypersensitivity to external noise or emotional stimuli. TaVNS directly attenuates autonomic nerve imbalance, activates cholinergic neuron system, and inhibits the release of inflammatory mediators. Imaging studies have shown that taVNS regulates neuronal activity in the gray matter, hypothalamus, thalamus, and hippocampus during the vertical phase, which are brain regions closely related to sleep disorders [113].

#### 3.2.5. Parasympathetic Control of Autonomic Functions

VNS activates the parasympathetic nervous system through the VN, which plays a central role in regulating autonomic functions. By modulating parasympathetic tone, VNS can restore autonomic balance, particularly in conditions characterized by sympathetic overactivity, such as heart failure, inflammatory diseases, and metabolic disorders. For instance, studies have demonstrated that VNS improves heart rate variability and reduces sympathetic hyperactivity in patients with heart failure, highlighting its direct influence on autonomic regulation [114]. Additionally, VNS has been shown to enhance gastrointestinal motility and reduce inflammation through parasympathetic pathways, further supporting its role in autonomic control [17]. Ignoring this aspect limits the comprehensive understanding of VNS’s therapeutic potential and its broader applications in autonomic dysfunction-related conditions.

## 4. Equipment and Parameter Selection

### 4.1. iVNS and nVNS

VNS, as a kind of electrical stimulation therapy, has a long-term promising application history. VNS usually refers to invasive vagus nerve stimulation (iVNS). With the deepening of theoretical research and the development of technology, people have developed non-invasive vagus nerve stimulation (nVNS) devices (Figure 2). In the past 20 years, both iVNS and nVNS have shown good results in the treatment of various diseases, including epilepsy, depression, and post-stroke motor rehabilitation.

iVNS refers to the surgical implantation of electrode devices to the left cervical VN under general anesthesia or local anesthesia and then fixing the stimulating device on the chest. This method ensures that the current provided by the stimulator can be directly transmitted to the VN, but there is a certain surgical risk, which may cause a series of complications due to hardware solidification and surgical operation, such as bradycardia, neck pain, wound infection, tracheal hematoma, and temporary vocal cord paralysis [115].

There are two ways to use nVNS devices: transcutaneous auricular VNS (taVNS) and transcutaneous cervical VNS (tcVNS). TaVNS refers to stimulating the ear nail position, and the current is transmitted to the ear branch nerves of VN through the skin. Yakunina and others reported that cymba conchae may be the best stimulation site [116]. TcVNS directly stimulates the cervical skin with a handheld device and transmits electrical signals to the cervical VN. fMRI studies showed that taVNS and tcVNS can produce similar effects as iVNS [117,118].

Closed-loop taVNS (CL-taVNS) was officially proposed in 2020 [119]. CL-taVNS is an automatic control taVNS system regulated by biofeedback signals: motor-activated au-ricular vagus nerve stimulation (MAAVNS) and respiratory-gated auricular vagal afferent nerve stimulation (RAVANS). MAAVNS is an EMG-gated CL-taVNS. MAAVNS pairs taVNS with sports activities and provides taVNS in targeted sports activities [120]. RAVANS works on the principle that inhalation induces transient inhibition of vagal nerve activity [121]. There are three possible CL-taVNS systems for the future: electro-encephalography (EEG)-gated CL-taVNS, electrocardiography (ECG)-gated CL-taVNS, and subcutaneous humoral signal (SHS)-gated CL-taVNS. These devices can be combined with artificial intelligence to facilitate clinical treatment, so as to play a greater role in the treatment of related diseases.

The controversy between iVNS and nVNS has not been fully discussed in existing studies. Although taVNS has attracted much attention due to its non-invasiveness and safety, its mechanism of action is still questionable. Some studies have shown that taVNS may indirectly affect brainstem nuclei such as the NTS and LC by activating the afferent fibers of the ABVN in the ear, thereby regulating autonomic nerve function and inflammatory pathways [116,117,118]. However, other studies have pointed out that the clinical effect of taVNS may be derived from the co-stimulation of adjacent nerves (such as trigeminal nerve or cervical plexus) or mediated by placebo effect, rather than specific activation of the vagal pathway [3,4]. For example, a number of fMRI studies have shown that taVNS stimulation of the earlobe (pseudo stimulation site) can still induce neural responses related to the deactivation of the auditory cortex, suggesting that its role may involve the mechanism of multisensory integration [116]. In addition, the existing taVNS clinical trials generally have methodological defects, such as a lack of double-blind controls, small sample sizes, inconsistent stimulus parameters (intensity, frequency, and duration), and no systematic evaluation of biomarkers of VN activation (such as heart rate variability or plasma acetylcholine level). Future studies need to use strictly designed sham stimulation control groups, standardized neuroimaging verification, and multimodal physiological indicators to clarify the target of taVNS and the similarities and differences between iVNS and nVNS. Only through more rigorous experimental design can we clarify the neuroregulatory mechanism of taVNS and promote its development to precision treatment.

In general, researchers who want to study the effect of VNS on certain diseases can first use iVNS or taVNS in animal models. If effective results are found, they can further consider using taVNS or tcVNS methods in clinical practice.

### 4.2. Parameters

Stimulation parameters have a significant impact on the efficacy of neuromodulation; however, the field generally lacks a consensus on optimum stimulation parameters [121]. We have listed the VNS methods and their parameters used in some representative research in Table 1.

When using VNS, the electrical pulse parameters that need attention are pulse width, frequency, and current intensity. The choice of current intensity and pulse width is particularly important. Hays found in animal experiments that moderate-intensity (0.8 mA) VNS best enhances motor cortex plasticity, while low-intensity (0.4 mA) and high-intensity (1.6 mA) VNS cannot enhance plasticity [122]. The plasticity of the motor cortex exhibits an inverted U-shape relationship with VNS intensity. This study suggests that VNS stimulation intensity can affect its stimulating effect. In addition, the interaction between current intensity and pulse width should be considered.

When the current intensity is equal, increasing the pulse width can increase the VNS effect [123]. However, in actual clinical applications, patients have different sensitivities to current intensity. Clinicians generally use current intensities within the range of values that are tolerable to patients. The current intensity of iVNS is generally between 0.25 and 3.75 mA, while the current intensity of taVNS and tcVNS depends on the manufacturer’s recommended parameters. The choice of pulse width is generally between 100 μs and 500 μs, and the frequency is generally set at 20–30 Hz. Because there is no unified parameter standard, researchers usually refer to the values reported in the research literature on similar diseases for reference. There is no standard for the number and duration of vagus nerve stimulation sessions. In general, when using iVNS, a continuous electrical stimulation of 30 s is given every 5 min. The specific number and duration of sessions are adjusted according to the experimenter’s experimental arrangements.

**Table 1 cimb-47-00122-t001:** VNS usage parameters in representative studies.

Disease	Type	Parameter				References
		Pulse Width	Frequency	Current	Time Administered	
Epilepsy	iVNS	500 μs	30 Hz	Tolerable setting (0.25–3.75 mA)	30 s ON, 5 min OFF, 14 weeks	Handforth et al. [49]
	taVNS	300 μs	10 Hz	Tolerable setting	3 times/1 day, 9 months	Hermann Stefan et al. [124]
Depression	iVNS	500 μs	20 Hz	Tolerable setting (0.25–3.5 mA)	30 s ON, 5 min OFF, 12 weeks	John Rush et al. [55]
	taVNS	200 μs	20 Hz	Tolerable setting (0–6 mA)	30 min each treatment, 2 times/1 day, 12 weeks	Peijing Rong et al. [125]
Headache	tcVNS	200 μs	25 Hz	Tolerable setting (≤60 mA)	120 s/dose, 6 dose/1 day, 8 weeks	Charly Gaul et al. [63] Alexander D Nesbitt et al. [126]
Stroke	iVNS	100 μs	30 Hz	0.8 mA (or lower)	0.5 s/dose, >300 dose per task, 3 times/1 week, 6 weeks	Jesse Dawson [68]
IBD	iVNS	500 μs	5 Hz	1 mA	10 s ON, 90 s OFF, 3 h/day, 5 days	J. Meregnani [71]
	iVNS	250 or 500 μs	10 Hz	0.25 mA~highest tolerable setting	30 s ON, 5 min OFF, 12 months	Valérie Sinniger [73]
RA	tcVNS	200 μs	25 Hz	Tolerable setting (≤60 mA)	120 s/dose, 3 doses per day, 5 days	AM Drewes [78]

Currently, it is still unclear whether there is a connection between VNS parameters and different mechanisms and diseases. In future studies, an exploration into the connection between VNS parameters and mechanisms, as well as the optimal parameter ranges for different diseases, should be considered. With the continued rapid growth in popularity and application of VNS, consensus research on optimal stimulation parameters may become one of the future directions for development.

## 5. Summary and Conclusions

On the one hand, VNS possesses the capability to activate innate “protective” mechanisms, encompassing anti-inflammatory effects, modulation of neurotransmitter release, enhancement of neural plasticity, reduction in BBB permeability, promotion of vascular regeneration, and inhibition of apoptosis and autophagy. These multifaceted effects have led to the official approval of VNS for clinical treatment of various conditions, including epilepsy, depression, headache, stroke, and obesity. Furthermore, its potential applications extend to anti-inflammatory treatment and the management of cardiovascular and cerebrovascular diseases, as well as other brain disorders.

On the other hand, the field of VNS still faces a series of challenges. Firstly, additional animal experiments are necessary to comprehensively elucidate the mechanism of action of VNS. These experiments should be complemented by clinical trials to further explore the potential applications of VNS. Secondly, a consensus on the optimal stimulation protocol for specific diseases remains elusive, which restricts its application in clinical settings. Therefore, future research on VNS will necessitate greater focus on addressing these issues.

## Figures and Tables

**Figure 1 cimb-47-00122-f001:**
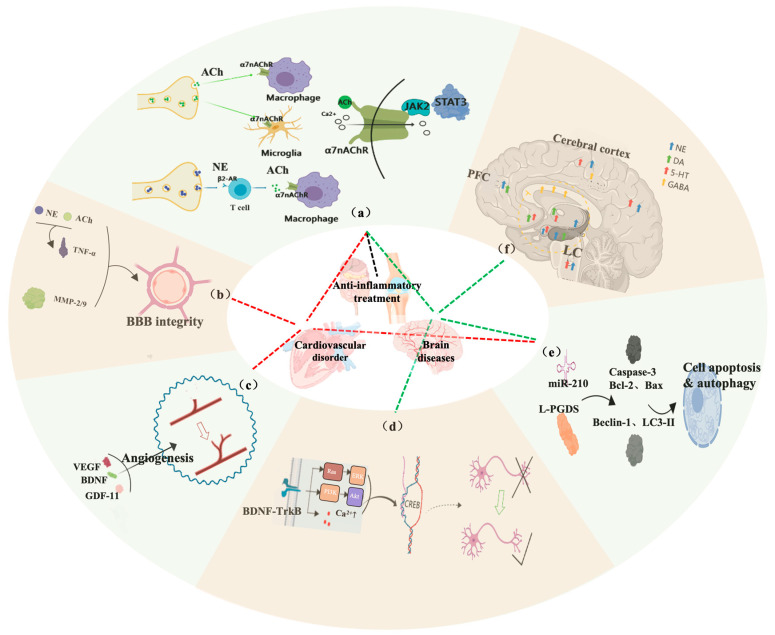
Potential mechanisms of VNS and its related clinical applications: (**a**) the VNS activates α7nAChR through three methods, which is an important basis for treating various diseases; (**b**) VNS regulates BBB integrity by modulating endothelial tight junction proteins; (**c**) VNS promotes angiogenesis by stimulating the release of VEGF, BDNF, and GDF-11, which serves as possible evidence for stroke rehabilitation; (**d**) VNS activates the BDNF-TrkB signaling pathway, promoting neural plasticity and neuroprotection; (**e**) VNS inhibits cell apoptosis and autophagy, reducing neuronal death and improving cell survival; and (**f**) VNS modulates the release of neurotransmitters in the brain, including ACh, NE, DA, 5-HT, and GABA, which is the mechanism for treating multiple brain diseases.

**Figure 2 cimb-47-00122-f002:**
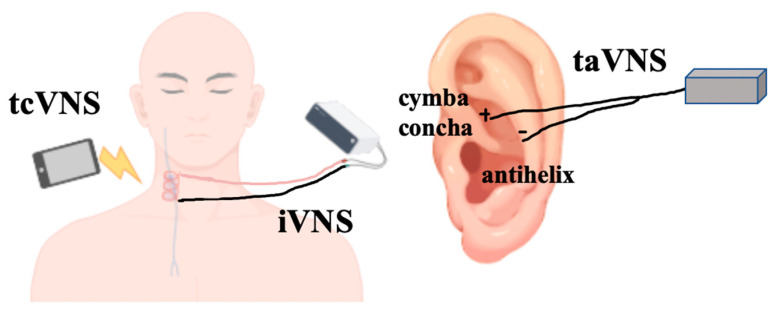
Three methods of VNS: iVNS typically involves surgical implantation of a cuff electrode on the right cervical VN; tcVNS utilizes a handheld stimulator to directly target the cervical region, where electrical signals are transmitted through the skin to the VN; taVNS focuses on stimulating specific auricular regions, such as the cymba concha and antihelix, which are innervated by the ABVN.

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
