# Peer review of "Mechanism and Applications of Vagus Nerve Stimulation"

_cimb, 2025, doi:10.3390/cimb47020122_

Round 1
Reviewer 1 Report
Comments and Suggestions for Authors
The manuscript by Chen Z. and Liu K. dives into some of the literature on the electrical stimulation of the vagus nerve and potential applications for clinical translation. The inclusion of equipment and parameters selection is a great compilation for the field. It appears that the goal of this article is to describe the mechanisms of action for VNS; however, authors fall short on this, and rather focus mostly on description of the applications. As a result, this manuscript is more a narrative review on applications than a review on mechanisms. Here are some of the suggestions that may strengthen this manuscript as a review on the mechanisms and applications of VNS:
1. All sections but mostly those on the applications focuses only on describing efficacy of the different therapies but make no mention of the hypothesized mechanisms of action. For example, authors do not describe what is the role of VNS on epilepsy or why it is believed to be efficacious.
2. In the potential uses, authors ignore the possibilities of direct parasympathetic control of autonomic functions.
3. The introduction is very small and does not provide enough anatomical and physiological information about the Vagus nerve; thus, it is hard to put its potential in context.
4. Authors are missing a complete discussion on the division that exists in the field between non-invasive and invasive stimulation of the vagus nerve. A lot of studies suggest that taVNS does not stimulate the vagus nerve and positive effects may be driven by different mechanisms. Most of the body of literature on taVNS requires more rigorous controls to demonstrate mechanism of action.
5. The introduction contains statements that are very subjective, including no definition of what authors refer to as "weak electrical currents".
6. Authors introduce James Corning as the first to introduce the concept of VNS, but do not describe the beginnings and/or rationale for its use.
7. Authors completely ignore that the mechanisms for neuroplasticity are related to the pairing of VNS with rehabilitative efforts and this section reads as if neuroplasticity just occurs spontaneously after VNS, which is not true.
8. Some acronyms are not defined. For example, VNS is not defined upon first appearance, but there are other instances.
9. Finally, it is not entirely clear what new information is synthetized in this review, especially when contrasted with recent literature reviews that focus in depth on each one of the applications described here (e.g., https://doi.org/10.3389/fnagi.2023.1173987)
Author Response
| 1. Summary
Thank you very much for taking the time to review this manuscript. For your suggestions, we have made a lot of changes in the text, such as adding the discussion of the possible relationship between some mechanisms and applications, citing some of the latest references (2020-2024), and modifying the position of each abbreviation for the first time. Please find the detailed responses below and the corresponding revisions/corrections highlighted/in track changes in the resubmitted files. 2. Point-by-point response to Comments and Suggestions for reviewerComments 1: All sections but mostly those on the applications focuses only on describing efficacy of the different therapies but make no mention of the hypothesized mechanisms of action. For example, authors do not describe what is the role of VNS on epilepsy or why it is believed to be efficacious. |
|
Response 1: Thank you for pointing this out. We agree with this comment. Therefore, we have added and described the possible biological mechanism of VNS as a treatment for specific disease. These contents can be found on line 255-259, 274-283, 292-303, 313-326, 337-359, 396-412, and line 464-469 in the revised manuscript. |
|
Comments 2: In the potential uses, authors ignore the possibilities of direct parasympathetic control of autonomic functions. |
|
Response 2: Agree. We did ignore the possibility of this approach. We have added relevant discussions in Section 3, which can be found in page 11, paragraph 3, and line 536-547. |
|
Comments 3: The introduction is very small and does not provide enough anatomical and physiological information about the Vagus nerve; thus, it is hard to put its potential in context. |
|
Response 3: Thank you for pointing this out. We have added information about the vagus nerve in the introduction, which you can find on lines 25-36. |
|
Comments 4: Authors are missing a complete discussion on the division that exists in the field between non-invasive and invasive stimulation of the vagus nerve. A lot of studies suggest that taVNS does not stimulate the vagus nerve and positive effects may be driven by different mechanisms. Most of the body of literature on taVNS requires more rigorous controls to demonstrate mechanism of action. |
|
Response 4: Thank you for pointing this out. We have added relevant discussions in lines 587-634, and clearly proposed that taVNS may cause different biological effects. |
|
Comments 5: The introduction contains statements that are very subjective, including no definition of what authors refer to as "weak electrical currents". |
|
Response 5: Agree. We have now revised the relevant contents of the second paragraph of the introduction to point out that VNS is the transmission of electrical signals to the vagus nerve. You can find it on lines 37-39. |
|
Comments 6: Authors introduce James Corning as the first to introduce the concept of VNS, but do not describe the beginnings and/or rationale for its use. |
|
Response 6: Thank you for pointing this out. We have reviewed James Corning's research on the first use of VNS in the treatment of epilepsy, and found that he was inspired by electrotherapy to put forward the concept of VNS at that time, and assumed that this kind of stimulation might suppress abnormal brain activity, reducing seizure frequency and severity. You can find it on lines 37-43. |
|
Comments 7: Authors completely ignore that the mechanisms for neuroplasticity are related to the pairing of VNS with rehabilitative efforts and this section reads as if neuroplasticity just occurs spontaneously after VNS, which is not true. |
|
Response 7: Thank you for pointing this out. We have added relevant discussions in lines 170-178 of the re-submitted review, and clearly put forward that the key to VNS promoting neural plasticity is to combine relevant rehabilitation training. |
|
Comments 8: Some acronyms are not defined. For example, VNS is not defined upon first appearance, but there are other instances. |
|
Response 8: Thank you for pointing this out. We have carefully examined the location of each abbreviation in the text and explained where it first appeared. You can also see the abbreviations we used on pages 14-15. |
|
Comments 9: Finally, it is not entirely clear what new information is synthetized in this review, especially when contrasted with recent literature reviews that focus in depth on each one of the applications described here (e.g., https://doi.org/10.3389/fnagi.2023.1173987) |
|
Response 9: Agree. We have updated the references and cited many articles published between 2020 and 2024. For example, you can find these newly cited references from the following places:Line 34, [4] Butt, M.F.; Albusoda, A.; Farmer, A.D.; Aziz, Q. The anatomical basis for transcutaneous auricular vagus nerve stimulation. J Anat 2020, 236, 588-611, doi:10.1111/joa.13122.
Line 36, [5] Jin, H.; Li, M.; Jeong, E.; Castro-Martinez, F.; Zuker, C.S. A body-brain circuit that regulates body inflammatory responses. Nature 2024, 630, 695-703, doi:10.1038/s41586-024-07469-y.
Line 68, [9] Fang, Y.T.; Lin, Y.T.; Tseng, W.L.; Tseng, P.; Hua, G.L.; Chao, Y.J.; Wu, Y.J. Neuroimmunomodulation of vagus nerve stimulation and the therapeutic implications. Front Aging Neurosci 2023, 15, 1173987, doi:10.3389/fnagi.2023.1173987.
Line 259, [48] Abdennadher, M.; Rohatgi, P.; Saxena, A. Vagus Nerve Stimulation Therapy in Epilepsy: An Overview of Technical and Surgical Method, Patient Selection, and Treatment Outcomes. Brain Sci 2024, 14, doi:10.3390/brainsci14070675.
Line 279, [52] Liu, C.; Tang, H.; Liu, C.; Ma, J.; Liu, G.; Niu, L.; Li, C.; Li, J. Transcutaneous auricular vagus nerve stimulation for post-stroke depression: A double-blind, randomized, placebo-controlled trial. J Affect Disord 2024, 354, 82-88, doi:10.1016/j.jad.2024.03.005.
Line 301, [58] Longo, S.; Rizza, S.; Federici, M. Microbiota-gut-brain axis: relationships among the vagus nerve, gut microbiota, obesity, and diabetes. Acta Diabetol 2023, 60, 1007-1017, doi:10.1007/s00592-023-02088-x. There are many more, so we won't list them one by one |

Reviewer 2 Report
Comments and Suggestions for Authors
Although there are multiple reviews on the subject, the article is interesting as an update since new advances in this area appear every day, however, the references worked in the article are very old, most of them between 2000-2015, the most recent is from 2021, so it is recommended to review the advances in the field, there are hundreds of articles published in recent years.
It is recommended to review the following articles published by MDPI that extensively review vagus nerve stimulation as therapeutic strategies in the treatment of various pathologies of the central nervous system and its effect on inflammation.
Role of the Cholinergic Anti-Inflammatory Reflex in Central Nervous System Diseases. Int J Mol Sci. 2021 Dec 14;22(24):13427, doi: 10.3390/ijms222413427
Vagus Nerve Stimulation Therapy in Epilepsy: An Overview of Technical and Surgical Method, Patient Selection, and Treatment Outcomes, Brain Sci. 2024, 14(7), 675; https://doi.org/10.3390/brainsci14070675
Pharmacological and Electroceutical Targeting of the Cholinergic Anti-Inflammatory Pathway in Autoimmune Diseases, Pharmaceuticals 2023, 16(8), 1089; https://doi.org/10.3390/ph16081089
Figure 1 has many elements and information, and the font is not very legible. As a suggestion, the size of the boxes on the left that contain the main information could be increased and the diagram of the human body could be reduced. You could also add information to the figure caption.
The article is a bit short compared to other current reviews, so it is recommended to address some topics in more depth.
Author Response
|
1. Summary |
|
|
||||||
|
Thank you very much for taking the time to review this manuscript. For your suggestions, we have made a lot of changes in the text, such as adding the discussion of the possible relationship between some mechanisms and applications, citing some of the latest references (2020-2024), modifying Figure 1 and adding Figure 2. Please find the detailed responses below and the corresponding revisions/corrections highlighted/in track changes in the re-submitted files.
|
||||||||

Reviewer 3 Report
Comments and Suggestions for Authors
The authors provide a comprehensive review of proposed mechanisms of vagal nerve stimulation and therapeutic applications. Suggestions:
1. Discuss: the subsets of taVNS (closed-loop): Respiratory-gated vagal afferent nerve stimulation (RAVANS) and Motor activated auricular VNS (MAAVNS)
2. Discuss the fascinating work on the use of taVNS in tinnitus (Yakunina et al Front Neurosci 2021)
3. There has been recent work on the possible benefit of VNS in Alzheimer's disease, which needs discussion (Slater et al. Clin Transl Med. 2021)
4. More discussion on the possible use of VNS to prevent postoperative ileus and systemic inflammatory response syndrome (SIRS) in the context of CAIP will be relevant (van Beekum et al. Auton Neurosc. 2021)
5. A brief mention of the potential use of taVNS in chronic insomnia will be desirable (Zhang et al. JAMA Netw Open 2024;7(12).
6. A figure showing the location of electrodes for iVNS, taVNS, and tcVNS may be helpful.
Author Response
|
1. Summary |
|
|
Thank you very much for taking the time to review this manuscript. For your suggestions, we have made a lot of changes in the text, such as discussing more meaningful topics, citing some of the latest references (2020-2024), adding Figure 2. Please find the detailed responses below and the corresponding revisions/corrections highlighted/in track changes in the re-submitted files.
|
|
|
2.Point-by-point response to Comments and Suggestions for Reviewer Comments 2: Discuss the fascinating work on the use of taVNS in tinnitus (Yakunina et al Front Neurosci 2021) |
|
|
Response 2: Thank you for pointing this out. We have added the fascinating work on the use of taVNS in tinnitus in page 11 and lines 518-527. Comments 3: There has been recent work on the possible benefit of VNS in Alzheimer's disease, which needs discussion (Slater et al. Clin Transl Med. 2021) Response 3: Agree. We have discussed the potential of VNS in treating Alzheimer's disease. You can find details on line 492-498. Comments 4: More discussion on the possible use of VNS to prevent postoperative ileus and systemic inflammatory response syndrome (SIRS) in the context of CAIP will be relevant (van Beekum et al. Auton Neurosc. 2021) Response 4: Thank you for pointing this out. We have discussed the possible use of VNS to prevent SIRS on line 386-394 in the resubmitted files. Comments 5: A brief mention of the potential use of taVNS in chronic insomnia will be desirable (Zhang et al. JAMA Netw Open 2024;7(12). Response 5: Thank you for pointing this out. We have discussed the potential use of taVNS in chronic insomnia on line 528-535 in the resubmitted files. Comments 6: A figure showing the location of electrodes for iVNS, taVNS, and tcVNS may be helpful. Response 6: We agree with this comment. Therefore, we have added Figure 2 to provide schematic diagrams of iVNS, tcVNS, and taVNS. You can see these changes in page 12. |
|

Round 2
Reviewer 2 Report
Comments and Suggestions for Authors
the authors have cited more recent articles and have taken into account the suggestions made. Thus, the article could be published in its present form, although I feel that the review is short and shallow compared to other current reviews, and that many advances and applications that have been implemented in recent years have been left out of the review.